# Glomerular developmental delay and proteinuria in the preterm neonatal rabbit

**Derek de Winter**[1]*, **Thomas Salaets**[2,3], **André Gie**[2], **Jan Deprest**[2,4,5], **Elena Levtchenko**[2,3], **Jaan Toelen**[2,3]

**1** Faculty of Medicine, Amsterdam UMC, Amsterdam, The Netherlands, **2** Department of Development and Regeneration, KU Leuven, Leuven, Belgium, **3** Department of Paediatrics, Division Woman and Child, University Hospitals Leuven, Leuven, Belgium, **4** Department of Obstetrics and Gynaecology, Division Woman and Child, University Hospitals Leuven, Leuven, Belgium, **5** Institute for Women's Health, University College London, London, United Kingdom

* derekdwinter@gmail.com

**Data Availability Statement:** All raw data is available from the Zenodo database (DOI: 10.5281/zenodo.3760389).

**Funding:** AG is funded with support of the Erasmus+ Programme of the European Union

## Abstract

Recent advances in neonatal care have improved the survival rate of those born premature. But prenatal conditions, premature birth and clinical interventions can lead to transient and permanent problems in these fragile patients. Premature birth (<36 gestational weeks) occurs during critical renal development and maturation. Some consequences have been observed but the exact pathophysiology is still not entirely known. This experimental animal study aims to investigate the effect of premature birth on postnatal nephrogenesis in premature neonatal rabbits compared to term rabbits of the same corrected age. We analyzed renal morphology, glomerular maturity and functional parameters (proteinuria and protein/creatinine ratio) in three cohorts of rabbit pups: preterm (G28), preterm at day 7 of life (G28 +7) and term at day 4 of life (G31+4). We found no significant differences in kidney volume and weight, and relative kidney volume between the cohorts. Nephrogenic zone width increased significantly over time when comparing G31 + 4 to G28. The renal corpuscle surface area, in the inner cortex and outer cortex, tended to decrease significantly after birth in both preterm and term groups. With regard to glomerular maturity, we found that the kidneys in the preterm cohorts were still in an immature state (presence of vesicles and capillary loop stage). Importantly, significant differences in proteinuria and protein/creatinine ratio were found. G28 + 7 showed increased proteinuria ($p = 0.019$) and an increased protein/creatinine ratio ($p = 0.023$) in comparison to G31 +4. In conclusion, these results suggest that the preterm rabbit kidney tends to linger in the immature glomerular stages and shows signs of a reduced renal functionality compared to the kidney born at term, which could in time lead to short- and long-term health consequences.

## Introduction

Recent advances in neonatal care have allowed the survival of extremely premature infants. At present more than 95% of infants born preterm survive into adulthood in most industrialized

(Framework Agreement number: 2013-0040). JD is funded by The Research Foundation – Flanders (FWO Flanders) as clinical researcher (1.801207). EL is funded by The Research Foundation – Flanders (FWO Flanders) as clinical investigator (1801110N). This study was supported by The Research Foundation – Flanders (FWO Flanders) (grant G0C4419N) and Katholieke Universiteit Leuven (C2 grant: C24/18/101). The funders had no role in study design, data collection and analysis, decision to publish, or preparation of the manuscript.

**Competing interests:** The authors have declared that no competing interests exist.

nations [1]. The prenatal conditions that cause preterm labor, the insult of preterm birth itself and the interventions during the clinical management of these fragile patients can all lead to transient and permanent effects on organ structure and function. This is often more evident for the pulmonary and nervous system compared to other organs, such as the kidneys.

Premature birth (<36 gestational weeks) affects the normal development and maturation of the renal system during a critical period. Nephrogenesis in humans is completed in week 34–36 of gestation, with a rapid formation of new nephrons from week 20 of gestation [2]. Preterm birth arrests this process and can result in a lower nephron number which does not recover later in life. Preterm birth itself is associated with neonatal acute kidney injury due to possible insults like decreased kidney perfusion (patent ductus, resuscitation, sepsis) or nephrotoxic medications. The resulting kidney damage is associated with renal pathology later in life such as hypertension, proteinuria and a decreased glomerular filtration rate [3–10].

Animal models are needed to study the exact pathogenesis or underlying mechanisms of kidney injury and disease caused by premature birth. Of the available models, mice and rats are most often used to study nephropathy, usually induced by surgical operation, drugs or toxins, or genetic modification. For neonatal pathology the rat seems an excellent model as active nephrogenesis persists until day 10 after birth. However, the rabbit model holds a position between the small and larger animal models and could thus provide a unique position for the translation of research to the clinic [11].

In this study we investigated the effect of preterm birth on nephrogenesis in preterm neonatal rabbits compared to term rabbits of the same corrected age.

## Materials and methods

### Ethical approval

The experiment was approved by the Ethics committee for Animal Experimentation of the Faculty of Medicine (KU Leuven; p060/2016) and performed according to current animal welfare guidelines and recommendations. Time-mated pregnant does (*New Zealand White and Dendermonde hybrid rabbits*) were provided by the animalium of the KU Leuven and housed in separate cages until caesarian section at a gestational age of 28 days (preterm) or 31 days (term). The cages were set at a 12 hour light-dark cycle, room temperature and free access to water and pellet feed.

### Experimental procedure

Prior to caesarian section, does were sedated with intramuscular ketamine 35 mg/kg bodyweight (Nimatek®; Eurovet Animal Health BV, Bladel, The Netherlands) and xylazin 6 mg/kg bodyweight (XYL-M®; VMD, Arendonk, Belgium). Does were placed in a supine position after adequate sedation and euthanized by administrating an intravenous injection of a mixture of 200 mg embutramide, 50 mg mebezonium, and 5 mg tetracain hydrochloride (T61®; Intervet International BV, Boxmeer, The Netherlands). Then, the uterus was rapidly exposed by creating an abdominal incision and all pups were delivered through hysterotomy.

From each mother allocated to preterm delivery (G28), the first 3 fetuses were randomly selected for fetal harvest. These fetal controls were prevented from breathing by not removing the fetal membranes. The remaining pups were dried, stimulated and put in an incubator at 32°C, 75% humidity, and 95% $O_2$. Term pups, all delivered through hysterotomy, were also put in an incubator at 32°C, 75% humidity, and 95% $O_2$ during the first hour after birth.

One-hour survivors were then weighed and labeled. All pups, term and preterm, were housed in an incubator at 32°C, 75% humidity, and 21% $O_2$. From then onwards, all pups were fed twice daily through a 3.5 Fr orogastric tube with milk replacer (Day One®, Protein

30%, Fat 50%; FoxValley; IL, USA) prepared according to manufacturer's guidelines. The quantity of feeding increased per postnatal day (PN) from 40 (PN day 0), to 50 (PN day 1), to 75 (PN day 2), and finally 100 mg/kg bodyweight on PN day 3 until harvest. Immunoglobins (Col-o-Cat®, SanoBest; Hertogenbosch, The Netherlands) were added during the first 2 PN days and probiotics, electrolytes and vitamins (Bio-Lapis®; Probiotics International Ltd.; Somerser, UK) were added during the first 5 PN days. A single intramuscular injection of vitamin K1 was administered on PN day 2 (0.002 mg/kg BW, Konakion pediatrique®; Roche, Basel, Switzerland). From PN day 2 onwards both term and preterm pups were given daily intramuscular injections of benzylpenicillin (20,000 I.U./kg BW Penicilline®; Kela, Sint-Niklaas, Belgium) and amikacin (20 mg/kg BW day, Amukin®; Bristol-Myers-Squibb, Brussels, Belgium).

## Tissue collecting and processing

Pups were deeply sedated with ketamine 35 mg/kg bodyweight (Nimatek®; Eurovet Animal Health BV, Bladel, The Netherlands) and xylazin 6 mg/kg bodyweight (XYL-M®; VMD, Arendonk, Belgium). After adequate sedation, the abdomen was opened through laparotomy and pups were euthanized by exsanguination. Blood from G28 + 7 and G31 + 4 pups was aspirated from the right ventricle, heparinized and centrifuged to retrieve plasma. Urine from G28 + 7 and G31 + 4 pups was aspirated directly from the bladder. Then, left kidneys were harvested and the perinephric fat was removed. Kidneys were weighed and fixated in PFA 4% for 24 hours. Then, kidney volume was measured by water displacement and the fixated kidneys were transferred to ethanol 70% for storage. Relative kidney volume and weight were calculated by dividing the kidney volume and weight by bodyweight at harvest.

One complete paraffin section at a thickness of 5 μm through the mid-hilar region on the longitudinal axis was collected for each kidney and stained with hematoxylin and eosin (H&E). Sections were digitalized using a Carl Zeiss Axio Scan.Z1 scanner at 40x magnification (0,220 μm$^2$ per pixel) and assessed blinded, by coding of the file names. Measurements were done by a single researcher using QuPath for Windows, Version 0.1.2 [12]. Unblinding was done after finishing measurements in all histology slides.

## Histological assessment

Kidney slides were digitally divided in equal poles (based on surface area) of which one pole was randomly selected (for more details see S1 Fig).

The distribution of maturity was quantified by counting and scoring all identifiable glomerular cross-sections in the selected half. Glomerular cross-sections unidentifiable for their maturity stage were excluded. Glomerular maturity stages were scored according to parameters described in Fig 1. Each stage was then averaged and calculated as the percentage of the total glomerular cross-sections counted.

Glomerular density—the number of glomerular cross-sections per surface area—was calculated by dividing the total glomerular cross-sections by the renal cortex surface area in mm$^2$. The renal cortex surface area was defined as the area between the medulla and the fibrous renal capsule and was measured by tracing these borders.

Nephrogenic zone width [μm] was defined as the area in the outer renal cortex just below the fibrous renal capsule to the last appearance of comma- and S-shaped bodies (S1 Fig) [16]. Histology slides were divided in eight equal sectors in which one measurement per sector, following the major organ surface area, was performed. Finally, the average nephrogenic zone width was calculated for each slide.

Renal corpuscle cross-sectional surface area [μm$^2$] was measured by tracing the inner lining of the Bowman's capsule. All glomerular cross-sections (stages I-III for the inner cortex, and

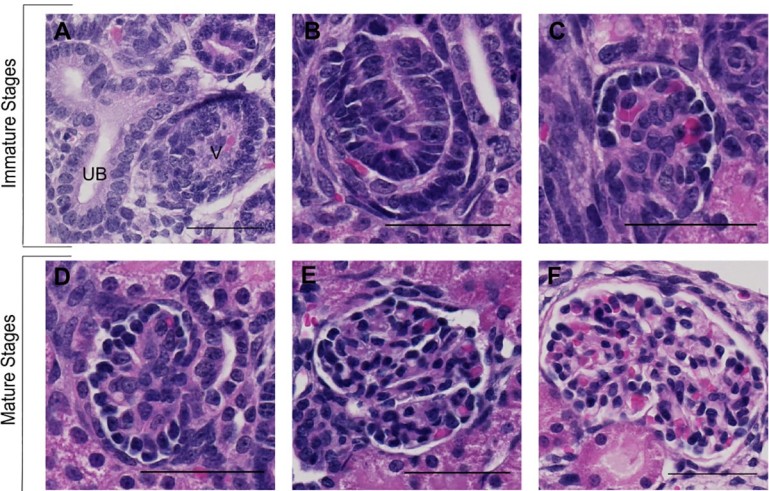

**Fig 1. Glomerular maturity stages found in the neonatal rabbit kidney.** Stages are based on criteria previously defined [13–15]. Glomeruli were divided in either immature stages (A-C) or mature stages (D-F). **(A)** Vesicle (V): a cluster of mesenchymal cells, exclusively found in the nephrogenic zone, bordering a ureteric bud (UB). **(B)** Comma- or S-shaped body: formed by the elongation and twisting of the vesicle. **(C)** Capillary loop: The lower limb of the S-shaped body formed an immature glomerulus, characterized by a crescent shape and few capillaries. **(D)** Stage I: no lobulation and at least half of the circumference is lined with dark-staining podocytes. **(E)** Stage II: little lobulation and less than half of the circumference is covered with dark-staining podocytes. Some open capillaries may be seen. **(F)** Stage III: fully matured glomerulus showing full lobulation and open capillaries. No podocytes lining the glomerular tuft. (*Bar = 50 μm*).

capillary loops for the outer cortex) were numbered. For both the inner and outer cortex 30 glomerular cross-sections or the maximum amount possible, were randomly selected using a random number generator. Then, the average renal corpuscle cross-sectional surface area for the inner and outer cortex was calculated.

## Functional parameters

Urinary creatinine [mg/dL] was determined through Roche enzymatic method, and urinary protein [g/L] with the benzethonium chloride method [17]. From this the protein/creatinine ratio was determined [g/g creatinine]. Plasma creatinine concentration [mg/dL] was determined by colorimetric assay (modified Jaffe reaction) [18]. Functional parameter data is presented as mean [CI-95%].

## Statistical analyses

Statistical analyses were performed using GraphPad Prism version 7.04 for Windows (GraphPad Software, La Jolla California USA, www.graphpad.com). A *p* value < 0.05 was considered statistically significant. All variables were assessed for normality using the Shapiro-Wilk test. Bodyweight, absolute and relative kidney volume and kidney weight, glomerular density, nephrogenic zone width, renal corpuscle cross-sectional surface area and the distribution of maturity were tested with a one-way analysis of variance (ANOVA) and a post hoc Tukey's multiple comparisons test. Functional data was tested with an independent samples T-test including Levene's test for equality of variances. Correlations on continuous variables were tested using the Spearman's Rank-Order correlation. Best-fit linear lines were plotted using linear regression analysis.

## Results

### Animal survival and biometry

A total of 19 pups were used in this experiment (S1 Table). Eight G28 pups were harvested directly after birth and six G28 pups were raised for harvesting on PN day 7 (G28 + 7). Six term pups were raised and harvested on PN day 4 (G31 + 4). Preterm or term pups that died before reaching their postnatal endpoint were excluded from the study (one G28 + 7 pup). This resulted in 8 G28, 5 G28 + 7 and 6 G31 + 4 pups. Birthweights of G28 and G28 + 7 were comparable. Even though harvest weight yielded no significant difference between G28 + 7 and G31 + 4, bodyweight tended to be higher in the term group ($p$ = 0.245).

### Kidney morphology

Relative kidney weight was found to be higher in G28 + 7 compared to G31 + 4 ($p$ = 0.023) (Fig 2). We found no significant differences in absolute kidney volume and weight, and relative kidney volume between G28 + 7 and G31 + 4, but in both term and preterm it tended to

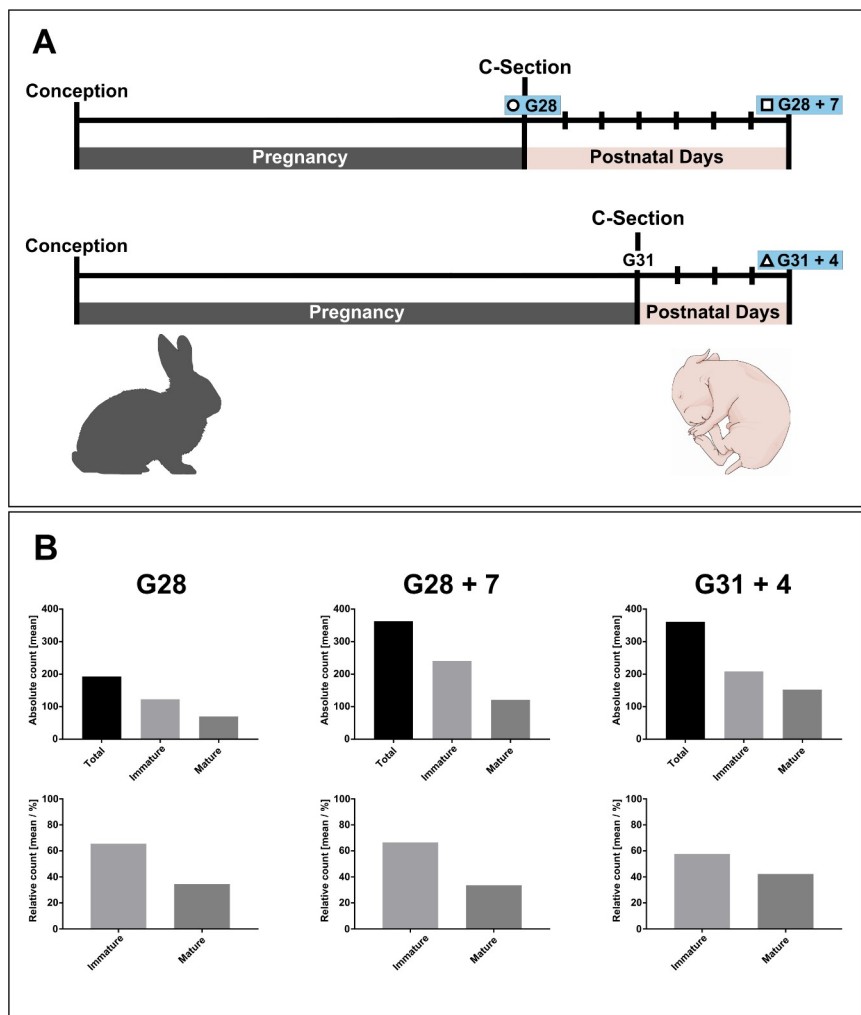

**Fig 2.** (A): Visual illustration of the animal model. Time points highlighted in blue are harvesting time points. (B): A graphic overview of the total immature and mature stage glomeruli across G28, G28 + 7 and G31 + 4, in absolute [mean] and relative counts [mean / %].

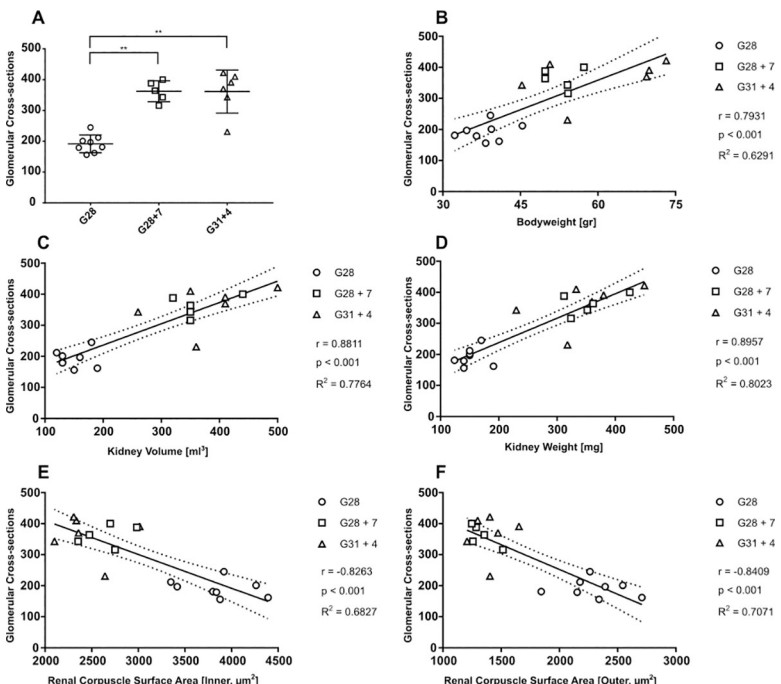

**Fig 3.** (A) Glomerular cross-sections counted. ($^{**}$ = p < 0.001. Error bars: mean ± SD) (B-F) Correlations plotted using Spearman's Rank-Order correlation on glomerular cross-sections and the following variables: bodyweight, kidney volume, kidney weight, and renal corpuscle surface area for the inner and outer cortex. Best-fit slope (solid line) is accompanied by a 95%-CI (dotted line).

increase compared to fetal controls. Moreover, we found that the average nephrogenic zone width increased overtime when comparing G31 + 4 to G28 ($p < 0.001$). But, this effect was not found when comparing G28 + 7 to G28. Nevertheless, the nephrogenic zone width was not significantly different between G28 + 7 and G31 + 4. The total glomerular cross-sectional count (Fig 3), renal cortex surface area (S1 Table) and glomerular density (Fig 2) tended to increase compared to fetal controls, but were comparable between G28 + 7 and G31 + 4. Post hoc Tukey's revealed no significant differences in renal corpuscle surface area between G28 + 7 and G31 + 4 (S1 Table). The renal corpuscle surface area, in the inner cortex and outer cortex, tended to decrease significantly ($p < 0.001$) after birth in both preterm and term groups. In addition, we found a negative correlation between the renal corpuscle surface area–for both the inner and outer cortex–and the glomerular cross-sectional count (Fig 3). Moreover, we found a positive correlation between the glomerular cross-sectional count and bodyweight (r = 0.7931), kidney volume (r = 0.8811) and kidney weight (r = 0.8957), indicating that the larger kidney cross-sections have more glomerular cross-sections.

## Glomerular maturity

Immature stages, taken overall, were found to be dominant in each group (Fig 2B and Table 1), specifically, vesicles in G28 (27.3%) and capillary loops in G28 + 7 (27.6%). Even though immature stages were dominant in G31 +4, we mostly found stage I glomeruli (31.0%). In comparison to fetal controls both G28 + 7 and G31 + 4 tended to have an increased absolute count of immature stages and stage I and II glomeruli. We found that G28 + 7 tended to have a higher relative count of overall immature glomerular stages than G31 + 4 ($p = 0.136$), and that G31 + 4 tended to have more mature glomeruli than G28 + 7. Furthermore, post hoc

**Table 1. Distribution of maturity stages.** Stages presented by the absolute mean, 95% confidence intervals and as the percentage of the total glomerular cross-sections counted.

| Stage | G28 | | | G28 + 7 | | | G31 + 4 | | |
|---|---|---|---|---|---|---|---|---|---|
| | Absolute | 95%-CI | Relative | Absolute | 95%-CI | Relative | Absolute | 95%-CI | Relative |
| **Immature Stages** | 122.9 | [96.9, 148.9] | 65.5% | 241.0** | [198.0, 284.0] | 66.5% | 208.5** | [170.9, 246.1] | 57.7% |
| Vesicle | 52.8 | [41.8, 63.8] | 27.3% | 79.2* | [64.5, 93.9] | 21.9%* | 60.0 | [41.1, 78.9] | 16.6%** |
| Comma or S-shaped Body | 33.5 | [29.0, 38.0] | 17.6% | 61.8** | [52.6, 71.0] | 17.1% | 69.3** | [59.6, 79.0] | 19.2% |
| Capillary Loop | 40.1 | [28.1, 52.1] | 20.6% | 100.0** | [69.8, 130.2] | 27.6% | 79.2* | [57.8, 100.6] | 21.9% |
| **Mature Stages** | 70.0 | [62.0, 78.0] | 34.5% | 121.2* | [97.1, 145.3] | 33.5% | 152.7** | [104.2, 201.2] | 42.3% |
| Stage I | 45.8 | [38.3, 53.3] | 24.0% | 88.6* | [68.2, 109.0] | 24.4% | 114.2** | [81.0, 147.4] | 31.0%* |
| Stage II | 16.4 | [12.6, 20.2] | 8.8% | 29.4 | [26.5, 32.3] | 8.1% | 36.5* | [16.6, 56.4] | 10.0% |
| Stage III | 3.1 | [1.3, 5.0] | 1.7% | 3.2 | [0.5, 5.9] | 0.9% | 2.0 | [0.4, 3.6] | 0.6% |

* $p < 0.05$ compared to G28.

** $p < 0.001$ compared to G28.

Tukey's yielded no significant differences between G28 + 7 and G31 + 4 in vesicles ($p = 0.064$) and stage I glomeruli ($p = 0.057$). No significant differences were found in the relative counts of comma or S-shaped bodies, capillary loops, stage II or stage III glomeruli.

## Functional parameters

We found a significant difference in urinary protein ($p = 0.019$) and protein/creatinine ratio ($p = 0.023$) between G28 + 7 and G31 +4 (Fig 4). No differences were found in urinary creatinine nor in plasma creatinine.

## Discussion

In humans preterm birth exacts a toll on many organ systems, especially the lung, brain and kidney. This results in an increased morbidity later in life with obstructive lung pathology, impaired neurocognitive outcomes and chronic renal disease with hypertension [3–10], but the pathophysiological background is still not entirely known. This is in part due to the lack of data from human subjects, because of ethical concerns and a large developmental variability [19], which forces researchers to rely on animal studies. In recent years we have developed a rabbit model to study the changes in pulmonary development that are induced by prematurity and hyperoxia. In the case of lung development, the rabbit mimics the human neonatal situation quite well as rabbits also reach the alveolar stage before birth. Also for the central nervous system we found that the rabbit was a good proxy for humans as they develop a form of encephalopathy of prematurity [20]. In order to further explore the translational value of the preterm rabbit model we set out to study the effect on renal development.

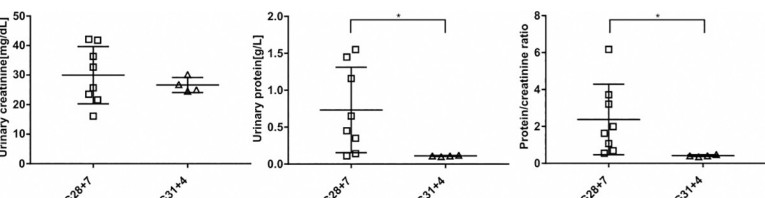

**Fig 4. Functional parameters.** Visual representation of the functional parameters measured: urinary creatinine, protein and the protein/creatinine ratio. Comparison between G28 + 7 and G31 + 4 reveals significant differences in proteinuria and in the protein/creatinine ratio. (* = p < 0.05. Error bars: mean ± SD).

We evaluated the effect of preterm birth on nephrogenesis in preterm neonatal rabbits compared to term rabbits of the same corrected age. For this we established three cohorts: G28, G28 + 7 and G31 + 4. This allowed us to study the effect of 3 additional intrauterine days by comparing G28 + 7 (28 intrauterine—and 7 extrauterine days) and gestational corrected G31 + 4 (31 intrauterine–and 4 extrauterine days) to the 'fetal preterm stage' at a gestation of 28 days. To our knowledge this is the first study to study the effects of prematurity on glomerular maturation and functionality.

When we analyzed the major structural aspects of the renal structures, we found that the kidney-to-bodyweight ratio was significantly higher in preterm rabbit pups (G28), which is in line with studies performed in human and non-human primate neonates [15,21,22]. But the difference of three days of intrauterine life did not result in major structural discrepancies between the two cohorts. The kidney-volume-to-bodyweight ratio and glomerular density did not differ between the G28 + 7 and G31 + 4 cohorts.

Yet the glomerular density as such may not be a sensitive enough marker to quantify differences between these two conditions. We performed a detailed assessment of the different stages of glomerular development and found out that the preterm rabbit kidney at a postnatal age of 7 days (G28 + 7) contains more immature glomerular structures (vesicles) compared with 'term' kidneys at a similar age (G31 + 4), which contain more stage I glomeruli. This suggests a significant disruption of normal nephrogenesis induced by prematurity. One can only speculate on the exact pathophysiological processes that would result in an altered nephrogenesis after birth in our model (e.g. water- and electrolyte disturbances, toxic effects of medication. . .).

In contrast to our findings, Sutherland et al. [15] found an accelerated maturation, with a decreased relative amount of vesicles and a smaller nephrogenic zone width in autopsied preterm human neonates whose post-natal survival ranged from 2 to 68 days, compared to still-born gestational controls. However, it remains unclear whether this difference was found because of the effect of prematurity or through the administration of antenatal steroids. Additionally, a study comparing preterm and term baboons found an interruption of normal kidney development, objectified through a decrease in glomerular generations and glomerular relative area with an increase of the renal corpuscle area [23].

It has been reported that nephrogenesis in the rabbit is completed in the 2nd or 3rd postnatal week [24]. In this respect the rabbit differs significantly from humans where the formation of new glomeruli end in the final weeks of gestation and no new structures are formed postnatally. But when neonates are born premature their kidneys are not yet fully developed and in that respect their ongoing nephrogenesis is similar to the preterm rabbit pups. In our experiments we confirm that nephrogenesis is indeed still ongoing after birth in the both the preterm and term cohorts by the presence of a nephrogenic zone and changes in total glomerular count, kidney volume and weight (compared to the fetal kidneys at G28). At present we cannot perform long term experiments to study further kidney maturation in preterm rabbits as the pups fail to survive beyond 10 days when fed artificially. So we cannot determine if this delayed nephrogenesis after preterm birth catches up and results in structurally and functionally normal mature kidneys. To our knowledge, the current literature provides no reference to what gestational age in humans corresponds to a gestational age of 28 days in the rabbit. We could argue that the term rabbit could be used as a 'preterm' model as we have proven that nephrogenesis continues after birth in the term rabbit. However, if term rabbits are used, the insult of premature birth itself and its effects would be minimized.

Next to the structural changes, we studied the functional differences between the two cohorts. It became clear that the kidneys of the preterm cohort suggest have a reduced functionality, shown by an increased urinary protein/creatinine ratio and higher levels of urinary

protein. Different hypotheses can be put forward to explain an increased protein/creatinine ratio in preterm rabbits. A study by Stelloh et al. has shown that preterm birth in mice correlates with a decrease in nephron endowment and thus induces stress on the remaining glomeruli, leading to proteinuria and hyperfiltration [25]. In our data, however, there is no compensatory increase in glomerular size, which is typically seen along with hyperfiltration, arguing against this theory. An alternative explanation are potential structural differences within the glomerulus. Wharram et al. have previously shown in a transgenic rat strain–in which podocytes express the human diphtheria toxin receptor—that a <40% podocyte reduction resulted in transient proteinuria and a reversible reduction of renal function. In contrast, a >40% podocyte reduction was responsible for segmental sclerosis, sustained proteinuria and a reduction in renal function [26]. In addition, Tsukahara et al. have described increasing and sustained high levels of albuminuria and urinary beta-2 microglobulin with increasing degrees of prematurity in human preterm neonates, whilst in term human neonates the protein levels gradually decreased during the first 28 days of life. Thus, demonstrating an increase in glomerular permeability and a decrease in tubular resorption with increasing degrees of prematurity [27]. However, Guignard JP. and Drukker A. found that in the preterm infant creatinine backflows along the tubule as a consequence of a 'leaky' immature tubular and vascular structures [28]. From this, we hypothesize that the increase in proteinuria in the absence of glomerular hypertrophy, found in preterm rabbits, may be caused by changes in the glomerular structure, such as podocyte immaturity or an "absolute" podocyte depletion (a decrease in the total number of healthy podocytes per glomerulus [29]) or through premature tubular processes such as tubular backflow of creatinine [28] and reduced tubular resorption [27].

This study is not without limitations. Firstly, an estimate of the total glomerular count (stereology based), was not obtained. We did, however, measure the glomerular density as an estimate for the amount of glomeruli per area. This measurement is thought to be susceptible to confounding factors such as section thickness and glomerular hypertrophy [29]. Also, the small number of animals in the experimental arms are likely to result in type II statistical errors. This argues a lack of significant differences where there in fact may be a biological effect and thus accepting a possibly false null hypothesis. To minimize the effects of these possible confounders we have consistently used the same section thickness and measured the cross-sectional renal corpuscle surface area in all cohorts. Also, in the current setting we were not able to perform the experiment using cationic ferritin enhanced-MRI, a novel and promising tool to quantify perfused glomeruli. This method, however, would not provide insight into differences in the developmental stages of the glomeruli that were found [30,31]. Secondarily, kidneys were embedded in paraffin: a method known to cause 40–50% shrinkage of soft tissues such as the kidney [32]. Furthermore, the prophylactic use of amikacin antibiotics in both preterm and term groups could have altered our results through nephrotoxicity [33]. However, Mingeot-Leclercq et al. have described amikacin to be nephrotoxic, after glomerular filtration, in the epithelial cells of the proximal tubules. Also, the use of benzylpenicillin and amikacin is a commonly used combination of antibiotics in clinical settings and provides us with a reflection. And, in previous experiments the rabbits were prone to develop dermal infections further substantiating the use of the current antibiotic regiment. Also, all experimental groups received these antibiotics in a dosage adjusted for bodyweight.

In conclusion, we have combined histology and functional parameters providing new insight in postnatal nephrogenesis in the preterm rabbit. We have shown that the preterm rabbit tends to linger in the immature glomerular stages compared to term. Importantly, preterm rabbits also show signs of proteinuria and a higher protein-to-creatinine ratio, suggestive of hyperfiltration which could in time lead to short- and long-term health consequences. Future

research could focus on the processes that are involved in these changes and in the use of the preterm rabbit for pharmacokinetic studies.

## Supporting information

**S1 Fig. Illustrative overview of the histology methods performed.** (A): Representative image of the nephrogenic zone as found in the outer renal cortex. The lower border of the nephrogenic zone is highlighted (white line). (B): Example histology slides for G28, G28 + 7 and G31 + 4. Provides a quick overview of the relative size. (C): Illustrative example of the analysis methodology. The kidney was divided in 8 sectors using *grid lines*. The upper or lower pole was randomly selected and all *glomerular cross-sections* were counted and staged. The *renal cortex surface* was measured by tracing just underneath the outer renal capsule and along the boundary between medulla and cortex. The *nephrogenic zone width* was measured in 8 sectors; the *renal corpuscle surface area* was measured in the 4 sectors in the randomly selected pole. (TIF)

**S1 Table. Baseline characteristics of units of analysis.** G28, G28 + 7, and G31 + 4, presented as *mean* and *95% confidence intervals*. (PDF)

## Acknowledgments

JD, EL and JT conceived the study; DDW, TS and AG collected the data; DDW analyzed the data, performed statistical analyses and DDW and TS drafted the manuscript; JD, EL and JT gave important intellectual contribution and critically revised the final manuscript.

## Author Contributions

**Conceptualization:** Jan Deprest, Elena Levtchenko, Jaan Toelen.

**Data curation:** Derek de Winter, Thomas Salaets, André Gie.

**Formal analysis:** Derek de Winter.

**Funding acquisition:** Jan Deprest, Jaan Toelen.

**Investigation:** Derek de Winter.

**Methodology:** Derek de Winter, André Gie, Elena Levtchenko.

**Project administration:** Derek de Winter.

**Supervision:** Jaan Toelen.

**Writing – original draft:** Derek de Winter, Thomas Salaets, Elena Levtchenko, Jaan Toelen.

**Writing – review & editing:** Derek de Winter, Thomas Salaets, Jan Deprest, Elena Levtchenko, Jaan Toelen.

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
