## [Decision Letter · Decision Letter 0]

24 Aug 2020

PONE-D-20-11602

Glomerular developmental delay and proteinuria in the preterm neonatal rabbit.

PLOS ONE

Dear Dr. de Winter,

Thank you for submitting your manuscript to PLOS ONE. After careful consideration, we feel that it has merit but does not fully meet PLOS ONE’s publication criteria as it currently stands. Therefore, we invite you to submit a revised version of the manuscript that addresses the points raised during the review process.

Please carefully address all the reviewers' remarks in your revised version of the manuscript.

We look forward to receiving your revised manuscript.

Kind regards,

Umberto Simeoni

Academic Editor

PLOS ONE

Journal Requirements:

2.At this time, we request that you  please report additional details in your Methods section regarding animal care, as per our editorial guidelines:

(1) Please provide additional details regarding the care of the rabbits prior to caesarian section.

(2) Please describe the care received by the rabbits after delivery, including the frequency of monitoring and the criteria used to assess animal health and well-being.

Thank you for your attention to these requests.

Reviewers' comments:

Reviewer's Responses to Questions

**Comments to the Author**

1. Is the manuscript technically sound, and do the data support the conclusions?

Reviewer #1: Yes

Reviewer #2: Partly

Reviewer #3: Yes

Reviewer #4: Yes

2. Has the statistical analysis been performed appropriately and rigorously? 

Reviewer #1: Yes

Reviewer #2: Yes

Reviewer #3: Yes

Reviewer #4: Yes

3. Have the authors made all data underlying the findings in their manuscript fully available?

Reviewer #1: Yes

Reviewer #2: Yes

Reviewer #3: Yes

Reviewer #4: Yes

4. Is the manuscript presented in an intelligible fashion and written in standard English?

Reviewer #1: Yes

Reviewer #2: Yes

Reviewer #3: Yes

Reviewer #4: Yes

5. Review Comments to the Author

Reviewer #1: no comments as i have a major conflict of interest with all the authors involved

no comments as i have a major conflict of interest with all the authors involved

no comments as i have a major conflict of interest with all the authors involved

no comments as i have a major conflict of interest with all the authors involved

Reviewer #2: The aim of this paper is to determine the effect of preterm birth on nephrogenesis of the rabbit, a proposed model for understanding the documented low nephron number in preterm human infants. In the Introduction, it is stated that “more than 95% of infants born preterm survive to adulthood” (line 48), but no reference is provided. Surely, this differs among populations.

The authors have based their study on 19 rabbits, 8 studied at preterm delivery, 6 born term and studied 7 days after birth, and 5 born preterm and studied 4 days after birth. The group G28+7 was raised in 95% oxygen and treated with amikacin. A sagittal section of the kidney was used for morphometric determination of relative glomerular number, glomerular size and maturation, and urine protein and creatinine concentration.

The results show that there was no difference between groups in kidney weight, kidney volume, or renal corpuscle surface area. Size of the nephrogenic zone was greater and proportion of immature glomeruli was lower in G31+4 vs. G28, but not different between G31+4 and G28+7. Urine protein concentration was increased (but with great variation) in G28+7 vs. G31+4 (with little variation). Table I is truncated, missing data for G31+4. An inverse relation was demonstrated between relative glomerular number and glomerular size.

The discussion admits that morphometric techniques utilized in the study may not be sensitive enough to detect differences between groups G31+4 and G28+7. This may explain the lack of increase in glomerular size in the G28+7 group demonstrated in the preterm mouse model cited in reference 24. P values between 0.05 and 0.15 suggest that type 2 error resulting from the small number of animals in each group (and high variation demonstrated in urine protein concentration in the G28+7 group) would support this. Stereologic approaches, including MRI image analysis, could circumvent this (Charlton JR et al. Pediatric Nephrology 2020 online publication https://doi.org/10.1007/s00467-020-04534-2). The MRI technique to study nephron number and structure has been employed in a rabbit model of neonatal acute kidney injury (Pediatric Research (2020) 87:1185–1192; https://doi.org/10.1038/s41390-019-0684-1).

Later study time points would also be highly desirable, but the authors indicate that these are not feasible in the model, because the preterm animals do not survive longer than a week with tube feeding. Additional concerns that limit the utility of the model are the exposure of the preterm neonatal rabbit to a nephrotoxin (amikacin) and a hyperoxic environment.

Despite the lack of documented glomerular hypertrophy, the authors account for increased urine protein concentration on the basis of hyperfiltration, citing the paper by Tsukahara (reference 26). However, albumin and beta-2-microglobulin were measured by Tsukahara et al., and changes in B2M were more pronounced than those in albumin, suggesting that tubular immaturity is a greater determinant than glomerular injury in preterm neonates. An additional factor to be considered is the backflow of creatinine across the immature renal tubule that artificially increases the urine protein/creatinine concentration in preterm urine (Guignard JP & Drukker A. Pediatrics, 1999 http://www.pediatrics.org/cgi/content/full/103/4/e49).

Reviewer #3: As you point out, human nephrogenesis is completed by 34-36 wk EGA. What gestational age in humans corresponds to d28 in the rabbit? Is this a model for previable human fetuses, or 28 wk ones?

Also, post-natal growth restriction is common in small premature infants. This corresponds with the fact that your 28+7 group weighed less than your 31+4 group. Since absolute kidney size was not different significantly, could the kidney-to-bodyweight ratio just reflect poor growth of the rest of the body, and not say anything about the kidney per se? I suspect providing adequate nutrition to these premie bunnies was very difficult, and like human ELBW, their somatic growth rate would not mimic that seen in utero.

Of note, on my copy, Table 1 was too wide for the page, and much of it was missing and could not be read.

Also, consider an additional reference with relevance to your discussion: Li J, et al. Nephrology 25 (2020) 116–124. doi: 10.1111/nep.13623 . They report that even though premies were small & lighter at follow up, kidney volume & length were not different from mature controls. How do your data fit with these human findings?

Keeping preterm rabbits alive if much trickier than term rabbits. Could term rabbits be used as a model for preterm human kidneys? Human nephrogenesis stops by 36 wks, but rabbits apparently continues postterm. Maybe term rabbit kidneys are a good model for premie human kidneys. Please comment on that.

I've made additional minor comments on the attached PDF Mark-up.

Reviewer #4: The authors of this manuscript investigated the effect of premature birth on nephrogenesis in preterm neonatal rabbits compared with term rabbits of similar corrected age.

This is an important neonatal topic in general, as emerging research focusing on the effects of prematurity on the kidney and its consequence for long-term renal outcome is being increasingly explored.

The manuscript was well-written and the study was thoughtfully designed. Nevertheless, there are some concerns with this manuscript.

Comments:

1. Animal models of kidney disease are reasonable to use to understand the pathogenesis of renal disease associated with the developmentally regulated changes in nephrogenesis and will aid in developing strategies to mitigate preterm kidney injury, however there are many inherent difficulties in bridging the gap from bench to bedside. The preterm rabbit model is an acceptable model as it the smallest model that includes a factor of prematurity and these animals are most likely to survive following surgical manipulation.

2. Translational research involving animals require careful experimental design, therefore the authors should explain why amikacin, a nephrotoxic agent, was used in the study design which aimed to describe normal development of the kidney with regards to glomerular development and renal function. Amikacin has been shown to increase creatinine level and the effect can persist for several days after cessation of therapy.

3. The authors should explain the respiratory state of those pups delivered at G28 and maintained for 7 days prior to being euthanized. It would seem intuitive that if there is a component of respiratory insufficiency, then hypoxia (a pathologic state) would be a contributory factor to the findings reported in these animals and do not adequately represent the real-life scenario where respiratory support and oxygen therapy may mitigate some of the changes described.

4. Authors should state whether the does are genetically identical – if not, how does that affect the model?

5. Authors should clarify how the pups from each mother was assigned to each arm of the study (that is equal distribution of pups from different mothers, especially between G28 and G28+7 arms of the study), because a similar in-utero environment may contribute to the ultimate developmental changes seen and may not represent the average effect seen across mothers and pups.

6. PLOS authors have the option to publish the peer review history of their article (what does this mean?). If published, this will include your full peer review and any attached files.

Reviewer #1: No

Reviewer #2: No

Reviewer #3: No

Reviewer #4: **Yes: **Janine Y Khan

---

## [Author Response · Author response to Decision Letter 0]

11 Sep 2020

Reviewer #1 provided no comments due to a major conflict of interest with all of the authors involved. 

Reviewer #2:

1. “The aim of this paper is to determine the effect of preterm birth on nephrogenesis of the rabbit, a proposed model for understanding the documented low nephron number in preterm human infants. In the Introduction, it is stated that “more than 95% of infants born preterm survive to adulthood” (line 48), but no reference is provided. Surely, this differs among populations.”

• We have provided a reference for the statement that more than 95% of infants born preterm survive into adulthood in most industrialized nations in line 48 of the Introduction. (Raju TNK, Pemberton VL, Saigal S, et al. Long-Term Healthcare Outcomes of Preterm Birth: An Executive Summary of a Conference Sponsored by the National Institutes of Health. J Pediatr. 2017;181:309-318.e1)

2. “The authors have based their study on 19 rabbits, 8 studied at preterm delivery, 6 born term and studied 7 days after birth, and 5 born preterm and studied 4 days after birth. The group G28+7 was raised in 95% oxygen and treated with amikacin. A sagittal section of the kidney was used for morphometric determination of relative glomerular number, glomerular size and maturation, and urine protein and creatinine concentration. The results show that there was no difference between groups in kidney weight, kidney volume, or renal corpuscle surface area. Size of the nephrogenic zone was greater and proportion of immature glomeruli was lower in G31+4 vs. G28, but not different between G31+4 and G28+7. Urine protein concentration was increased (but with great variation) in G28+7 vs. G31+4 (with little variation). Table I is truncated, missing data for G31+4. An inverse relation was demonstrated between relative glomerular number and glomerular size.”

• In fact, both G28 + 7 and G31 + 4 were raised in an incubator in 21% oxygen (normoxic) and were treated with amikacin from postnatal day 2 onwards. Preterm and term pups were housed at 95% oxygen during the first hour after birth only. This might not have been completely clear from the original Methods description. In the revised manuscript we provide a more detailed description.

• We apologize for Table 1 being truncated in the current version, as has been noted by multiple reviewers. Table 1 has been adapted to fit in ‘Landscape’ view, hopefully this will solve the issue and provide you with all the relevant information. 

3. “The discussion admits that morphometric techniques utilized in the study may not be sensitive enough to detect differences between groups G31+4 and G28+7. This may explain the lack of increase in glomerular size in the G28+7 group demonstrated in the preterm mouse model cited in reference 24. P values between 0.05 and 0.15 suggest that type 2 error resulting from the small number of animals in each group (and high variation demonstrated in urine protein concentration in the G28+7 group) would support this. Stereologic approaches, including MRI image analysis, could circumvent this (Charlton JR et al. Pediatric Nephrology 2020 online publication https://doi.org/10.1007/s00467-020-04534-2). The MRI technique to study nephron number and structure has been employed in a rabbit model of neonatal acute kidney injury (Pediatric Research (2020) 87:1185–1192; https://doi.org/10.1038/s41390-019-0684-1).”

• Indeed, morphometric techniques that have been used in this experiment are sensitive to confounding. We have tried to minimize these effects by consistently using the same section thickness and measuring cross-sectional renal corpuscle surface area in all cohorts. Stereologic approaches could indeed be more sensitive to detect more subtle changes between groups as would larger cohorts. For the latter we are bound by the European and Belgian legislation and ethical restrictions on animal research which do not allow large scale experiments. As there was very little published literature on this topic to guide a power calculation for the outcome parameters, we could not accurately predict the necessary sample size. 

• MRI image analysis as described by Charlton et al. provides a novel and promising tool to quantify perfused glomeruli and might be of great use in the future. In the current setting we were unable to perform the experiment using MRI image analysis. We have altered the manuscript in ‘Discussion’ and described MRI-analysis as a promising alternative.

4. “Later study time points would also be highly desirable, but the authors indicate that these are not feasible in the model, because the preterm animals do not survive longer than a week with tube feeding. Additional concerns that limit the utility of the model are the exposure of the preterm neonatal rabbit to a nephrotoxin (amikacin) and a hyperoxic environment.”

• Indeed, later study time points would be preferable as this would allow us to determine whether the glomerular developmental delay and proteinuria in the preterm rabbit are persistent or whether these changes resolve over time. At present, we are working on experiments to prolong the rabbit pup survival using different artificial milk substitutes, some of which show promising results.

• The rabbit pups were treated with amikacin from postnatal day 2 onwards in order to prevent infectious complications in these fragile pups. In previous experiments performed in our research group the rabbit pups were prone to dermal infections, and thus we chose to provide all experimental groups with prophylactic antibiotics. Mingeot-Leclercq et al. have described amikacin to be toxic to the epithelial lining of the proximal tubules, after glomerular filtration (and as such have an effect on kidney development/maturation). In a clinical context, an antibiotic regime of benzylpenicillin and amikacin is a often used in current neonatal care. As all the groups in our experiments receive this combination, all possible effects of amikacin would be equally present in all groups.

• The housing of the pups might not have been clearly described in the Methods section. In order to clarify this for the readers we have adapted the Method section. Briefly, all term and preterm pups were housed in an incubator at 95% O2 during the first hour after birth. Afterwards all one-hour survivors were housed in an incubator at 21% O2. We do appreciate that hyperoxia is no longer the most important risk factor in the etiology of BPD or other neonatal conditions, but it is an integral part of the model to mimic the lung injury. It is in this context that we intended to investigate the nephrological changes, not merely based on prematurity.

5. “Despite the lack of documented glomerular hypertrophy, the authors account for increased urine protein concentration on the basis of hyperfiltration, citing the paper by Tsukahara (reference 26). However, albumin and beta-2-microglobulin were measured by Tsukahara et al., and changes in B2M were more pronounced than those in albumin, suggesting that tubular immaturity is a greater determinant than glomerular injury in preterm neonates. An additional factor to be considered is the backflow of creatinine across the immature renal tubule that artificially increases the urine protein/creatinine concentration in preterm urine (Guignard JP & Drukker A. Pediatrics, 1999 http://www.pediatrics.org/cgi/content/full/103/4/e49).”

• Unfortunately we have not been able to perform specific protein analyses on albumin and beta-2-microglobulin in the current setting. It is certainly a factor, as a determinant for the origin of the proteinuria, to take into account in further research. 

• Guignard JP & Drukker A. have shown that the preterm neonatal infant shows tubular resorption of creatinine through backflow of creatinine along ‘leaky’ and immature tubules. In the current experiment we were unfortunately not able to determine whether this was the case in our preterm rabbits. We have thus adapted the Discussion accordingly and hypothesize that the difference in protein/creatinine ratio found in the preterm rabbit may be caused by changes in glomerular structure or through tubular backflow of creatinine. 

Reviewer #3:

1. “As you point out, human nephrogenesis is completed by 34-36 wk EGA. What gestational age in humans corresponds to d28 in the rabbit? Is this a model for previable human fetuses, or 28 wk ones? Also, post-natal growth restriction is common in small premature infants. This corresponds with the fact that your 28+7 group weighed less than your 31+4 group. Since absolute kidney size was not different significantly, could the kidney-to-bodyweight ratio just reflect poor growth of the rest of the body, and not say anything about the kidney per se? I suspect providing adequate nutrition to these premie bunnies was very difficult, and like human ELBW, their somatic growth rate would not mimic that seen in utero.”

• To our knowledge, the current literature provides no exact reference as to what gestational age in humans corresponds to a gestational age in rabbits as a whole. It is often provided for lung development (where it indeed corresponds to the canalicular / early glandular stage) yet there are no detailed data on comparative kidney development. We have adapted the Discussion accordingly to provide the reader with more insight in the transferability of the neonatal rabbit kidney model.

• Yes, the difference found in kidney-to-bodyweight ratio between the G31 + 4 and G28 + 7 experimental groups can certainly be explained by post-natal growth restriction that is commonly seen in both preterm human infants as well as in animal studies that study prematurity. Also in rabbits ex-utero growth of the premature pups does not mimic the growth seen in utero, as can be seen in Supplementary Table 1. G28 + 7 tended to have a lower harvest weight compared to G31 + 4. We have previously used foster animals (only possible in normoxia settings) and found that compared to gavage fed animals, the mother fed animals gained significantly more weight. So our model definitely involves a ‘failure to thrive’ postnatally.

2. “Of note, on my copy, Table 1 was too wide for the page, and much of it was missing and could not be read.”

• Indeed, Table 1 appears to be truncated in the current version of the manuscript. We have adapted this to be in ‘Landscape’ view.

3. “Also, consider an additional reference with relevance to your discussion: Li J, et al. Nephrology 25 (2020) 116–124. doi: 10.1111/nep.13623 . They report that even though premies were small & lighter at follow up, kidney volume & length were not different from mature controls. How do your data fit with these human findings?”

• Li J et al. have performed a retrospective study that compares the renal ultrasonographic ultrasounds between preterm infants and term infants. They compared ultrasound images at 32 weeks (preterm), 37 weeks and at 6 months of age. They measured kidney volume, length and also renal cortex and medulla thickness. They found that all kidney parameters were smaller compared with term babies, however by 6 months of age kidney volume and length were no longer significantly different between preterm and term infants. The catch-up growth was mainly attributable to hypertrophic growth of the renal cortex, whilst the renal medulla growth was impaired. 

In respect to our data we found no significant differences in renal cortex surface area of G28 +7 compared to G31 + 4. Of note, however, nephrogenesis was still ongoing in both experimental arms, whilst in the study by Li J et al. nephrogenesis was finished in the term infants as well as in the follow-up period. To adequately compare the rabbit to the human data we would need to extend the experimental time frame. 

4. “Keeping preterm rabbits alive is much trickier than term rabbits. Could term rabbits be used as a model for preterm human kidneys? Human nephrogenesis stops by 36 wks, but rabbits apparently continues postterm. Maybe term rabbit kidneys are a good model for premie human kidneys. Please comment on that.”

• This is an excellent suggestion. Indeed, term rabbits, in which nephrogenesis has been shown to continue postterm, could theoretically be used as a model for preterm human kidneys. However, a significant factor to take into consideration (and a factor that would be absent if term rabbits were used), is the changes resulting because of preterm birth itself (intrauterine development and its limited exposure to stressors and toxins is significantly different from postnatal development). 

5. “Reviewer #3 made additional minor comments on the attached PDF Mark-up.”

• We have taken the comments in the attached PDF Mark-up under consideration and have adapted the manuscript where needed.

• We have adapted the text in the section ‘Glomerular Maturity’ to clarify the results of the overall immature and mature glomerular stages in the experimental groups.

• Also, we have altered Table 1 to ‘Landscape’. Hopefully, this will resolve these issues.

• Lastly, Mingeot-Leclercq et al. have indeed described amikacin to be nephrotoxic, after glomerular filtration, in the epithelial cells of the proximal tubules and thus limits the negative effect of amikacin on our experiment that studies the glomerular development. We have adapted the ‘Discussion’ based on this information.

Reviewer #4:

1. “Animal models of kidney disease are reasonable to use to understand the pathogenesis of renal disease associated with the developmentally regulated changes in nephrogenesis and will aid in developing strategies to mitigate preterm kidney injury, however there are many inherent difficulties in bridging the gap from bench to bedside. The preterm rabbit model is an acceptable model as it the smallest model that includes a factor of prematurity and these animals are most likely to survive following surgical manipulation.”

• Indeed, the rabbit provides an experimental model that includes prematurity in a relatively small species. In current neonatal research data from human subjects are lacking and thus we often rely on animal studies (and the availability of detailed histological analysis). Through this way we hope to bridge the gap from bench to bedside.

2. “Translational research involving animals require careful experimental design, therefore the authors should explain why amikacin, a nephrotoxic agent, was used in the study design which aimed to describe normal development of the kidney with regards to glomerular development and renal function. Amikacin has been shown to increase creatinine level and the effect can persist for several days after cessation of therapy.”

• Indeed, pups were treated with amikacin, an antibiotic known to be nephrotoxic. This is definitely a limitation of the model, however the decision to use a regiment of benzylpenicillin and amikacin was based on previous experiments in which the rabbit pups were prone to develop severe dermal infections (with even increased mortality). Benzylpenicillin/amikacin is a commonly used combination of antibiotics in clinical settings and thus may represent in this context the current clinic scenario (yet we do not claim that all preterm infants receive this drug regimen). Also, all experimental groups (G28 +7 and G31+4) were treated with a dosage adjusted for body weight. On top of that, Mingeot-Leclercq et al. have previously described amikacin to be nephrotoxic in the epithelial cells of the proximal tubules, after glomerular filtration. In order to provide the reader with more insight in the use of amikacin and its limitations to our experiment we have altered the Discussion.

3. “The authors should explain the respiratory state of those pups delivered at G28 and maintained for 7 days prior to being euthanized. It would seem intuitive that if there is a component of respiratory insufficiency, then hypoxia (a pathologic state) would be a contributory factor to the findings reported in these animals and do not adequately represent the real-life scenario where respiratory support and oxygen therapy may mitigate some of the changes described.”

• This might not have been completely clear from the ‘Methods’ section. We have adapted this section to provide more clarity on this part of the experimental procedure. Briefly, one-hour survivors (both term and preterm) were housed in an incubator at 21% O2 until euthanisation. The animals are never subjected to an episode of hypoxia (only the feeding moments are under normoxic conditions, but these are very limited in time – measured in seconds). 

4. “Authors should state whether the does are genetically identical – if not, how does that affect the model?”

• In this model the does were not genetically identical, as is often the case in inbred mouse/rat models. This comes with both its advantages as well as drawbacks. The fact that the does were genetically diverse could prove the results to be less consistent than if the does were to be genetically identical, however it does provide a finer reflection of the clinical setting: a genetically diverse setting. 

5. “Authors should clarify how the pups from each mother was assigned to each arm of the study (that is equal distribution of pups from different mothers, especially between G28 and G28+7 arms of the study), because a similar in-utero environment may contribute to the ultimate developmental changes seen and may not represent the average effect seen across mothers and pups.”

• From each mother allocated to preterm delivery, at a gestational age of 28 days, the first 3 fetuses were randomly selected for fetal harvest. This was based on the order that the pups were delivered through hysterotomy. Meaning that the pups from the experimental arm G28 and the experimental arm G28 + 7 originated from the same does.

---

## [Decision Letter · Decision Letter 1]

8 Oct 2020

PONE-D-20-11602R1

Glomerular developmental delay and proteinuria in the preterm neonatal rabbit.

PLOS ONE

Dear Dr. de Winter,

After careful consideration, we invite you to perform an additional, minor's revision of your manuscript.

Indeed, reviewer No 2, while noting that many of his/her concerns have been addressed in your former revision, still has made two remarks that need to be taken into account in your final manuscript. 

We look forward to receiving your revised manuscript.

Kind regards,

Umberto Simeoni

Academic Editor

PLOS ONE

Reviewers' comments:

Reviewer's Responses to Questions

**Comments to the Author**

1. If the authors have adequately addressed your comments raised in a previous round of review and you feel that this manuscript is now acceptable for publication, you may indicate that here to bypass the “Comments to the Author” section, enter your conflict of interest statement in the “Confidential to Editor” section, and submit your "Accept" recommendation.

Reviewer #2: (No Response)

2. Is the manuscript technically sound, and do the data support the conclusions?

Reviewer #2: Partly

3. Has the statistical analysis been performed appropriately and rigorously? 

Reviewer #2: Yes

4. Have the authors made all data underlying the findings in their manuscript fully available?

Reviewer #2: Yes

5. Is the manuscript presented in an intelligible fashion and written in standard English?

Reviewer #2: Yes

6. Review Comments to the Author

Reviewer #2: 2 concerns remain:

1. The small number of animals in each group likely resulted in type 2 error in the statistical analyses, and in a lack of significant differences where there may be biologic effects. This should be indicated in the discussion. The comment in the author's response to reviewers, "morphometric techniques that have been used in this experiment are sensitive to confounding" would be magnified by the small number of animals.

2. In the discussion, lines 344-348, the authors hypothesize that increased proteinuria may be caused by podocyte pathology or tubular backflow of creatinine, but do not mention reduced tubular protein reabsorption that was proposed by Tsukahara based on their beta-2 microglobulin data. This should be included.

7. PLOS authors have the option to publish the peer review history of their article (what does this mean?). If published, this will include your full peer review and any attached files.

Reviewer #2: No

---

## [Author Response · Author response to Decision Letter 1]

9 Oct 2020

Reviewer #2:

1. “The small number of animals in each group likely resulted in type 2 error in the statistical analyses, and in a lack of significant differences where there may be biologic effects. This should be indicated in the discussion. The comment in the author's response to reviewers, "morphometric techniques that have been used in this experiment are sensitive to confounding" would be magnified by the small number of animals.”

• Indeed, the small number of animals in each experimental arm make it possible to create a type II statistical error, thus accepting a possibly false null hypothesis. We have adapted the manuscript in accordance and can be referenced on lines 355-357.

2. “In the discussion, lines 344-348, the authors hypothesize that increased proteinuria may be caused by podocyte pathology or tubular backflow of creatinine, but do not mention reduced tubular protein reabsorption that was proposed by Tsukahara based on their beta-2 microglobulin data. This should be included.”

• The reviewer is right. Tsukahara et al. have evaluated the presence of albumin and beta-2 microglobulin in urine samples during the neonatal period. They have found that both albumin and beta-2 microglobulin were increasingly elevated with increasing degrees of prematurity. In fact, changes in beta-2 microglobulin were more noticeable in relation to gestational age. Thus, we have adapted the manuscript to include this explanation. The data and findings from Tsukahara et al. are described in lines 337-342.

• In addition, we have adapted our hypothesis to fit the abovementioned findings by Tsukahara et al. The hypothesis is described from lines 344-349: “From this, we hypothesize that the increase in proteinuria in the absence of glomerular hypertrophy, found in preterm rabbits, may be caused by changes in the glomerular structure, such as podocyte immaturity or an “absolute” podocyte depletion (a decrease in the total number of healthy podocytes per glomerulus [29]) or through premature tubular processes such as tubular backflow of creatinine [28] and reduced tubular resorption [27].”

---

## [Decision Letter · Decision Letter 2]

14 Oct 2020

Glomerular developmental delay and proteinuria in the preterm neonatal rabbit.

PONE-D-20-11602R2

Dear Dr. de Winter,

We’re pleased to inform you that your manuscript has been judged scientifically suitable for publication and will be formally accepted for publication once it meets all outstanding technical requirements.

Kind regards,

Umberto Simeoni

Academic Editor

PLOS ONE

Additional Editor Comments (optional):

Reviewers' comments:

Reviewer's Responses to Questions

**Comments to the Author**

1. If the authors have adequately addressed your comments raised in a previous round of review and you feel that this manuscript is now acceptable for publication, you may indicate that here to bypass the “Comments to the Author” section, enter your conflict of interest statement in the “Confidential to Editor” section, and submit your "Accept" recommendation.

Reviewer #2: All comments have been addressed

2. Is the manuscript technically sound, and do the data support the conclusions?

Reviewer #2: (No Response)

3. Has the statistical analysis been performed appropriately and rigorously? 

Reviewer #2: (No Response)

4. Have the authors made all data underlying the findings in their manuscript fully available?

Reviewer #2: (No Response)

5. Is the manuscript presented in an intelligible fashion and written in standard English?

Reviewer #2: (No Response)

6. Review Comments to the Author

Reviewer #2: (No Response)

7. PLOS authors have the option to publish the peer review history of their article (what does this mean?). If published, this will include your full peer review and any attached files.

Reviewer #2: No

---

## [Editor Report · Acceptance letter]

19 Oct 2020

PONE-D-20-11602R2 

Glomerular developmental delay and proteinuria in the preterm neonatal rabbit. 

Dear Dr. de Winter:

I'm pleased to inform you that your manuscript has been deemed suitable for publication in PLOS ONE. Congratulations! Your manuscript is now with our production department. 

Kind regards, 

on behalf of

Dr. Umberto Simeoni 

Academic Editor

PLOS ONE